# Effective Locations for Injecting Botulinum Toxin into the Mentalis Muscle; Cadaveric and Ultrasonographic Study

**DOI:** 10.3390/toxins13020096

**Published:** 2021-01-27

**Authors:** Da-Yae Choi, Hyungkyu Bae, Jung-Hee Bae, Hee-Jin Kim, Kyung-Seok Hu

**Affiliations:** 1Department of Dental Hygiene, Catholic Kwandong University, 24 Beomil-ro 579beon-gil, Gangneung 25601, Korea; choi9989@cku.ac.kr; 2Department of Oral Biology, Division in Anatomy and Developmental Biology, Human Identification Research Institute, BK21 PLUS Project, Yonsei University College of Dentistry, 50-1 Yonsei-ro, Seodaemun-gu, Seoul 03722, Korea; hkbae410@yuhs.ac (H.B.); hjk776@yuhs.ac (H.-J.K.); 3Department of Dental Hygiene, Division of Health Sciences, Namseoul University, 91 Daehak-ro, Seobuk-gu, Cheonan 31020, Korea; jung18342@naver.com; 4Department of Materials Science & Engineering, College of Engineering, Yonsei University Seoul, 50 Yonsei-ro, Seodaemun-gu, Seoul 03722, Korea

**Keywords:** mentalis muscle, facial rejuvenating, botulinum toxin, injection procedure, facial rhytides, mental crease

## Abstract

The mentalis muscle is now considered key structures when performing procedures for rejuvenating the lower face. The aim of this study was to determine the anatomical morphology and location of the mentalis muscle and thereby provide anatomical information for facilitating clinical procedures designed to rejuvenate the lower face. Forty-four adult hemifaces from five Thai cadavers and 21 Korean cadavers were dissected to identify the locations of the mentalis muscle. Sixty-six hemifaces from 33 healthy young Korean subjects were included in an ultrasonographic study. The depth of the mentalis muscle below the skin surface, the thickness of the mentalis muscle, and the distance from the bone to the mentalis muscle were measured at the two points that were 5 mm lateral to the most-prominent point of the chin. The mentalis muscle was classified into two types based to its shape: in type A (86.4%, 38 of the 44 cases) it was dome shaped in three dimensions, while in type B (13.6%, 6 of the 44 cases) it was flat. The mentalis muscle was present mostly at the area 5–10 mm from the midsagittal line and 20–30 mm from a horizontal line connecting the mouth corners. The mentalis muscle was present between depths of 6.7 to 10.7 mm below the skin. This new information about the location of the mentalis muscle may help when identifying the most effective and safe botulinum toxin injection points and depths during esthetic procedures for weakened facial rhytides on the lower face.

## 1. Introduction

The mentalis muscle is paired with dome-shaped muscles of the central lower lip. It originates from the anterior mandible at the level below the attached gingiva and is attached to the skin of the chin. Usually these two muscles are separated, and some fat may lie between their separate origins [1,2,3]. The mentalis muscle elevates soft tissue of the mentum area and supports the proper position and movement of the lower lip. Problems with these functions can cause the lower central incisors to be visible in the resting position, or impair denture stability [1,3,4].

The mentalis muscle and its surrounding soft tissue (i.e., fat) are now considered key structures when performing procedures for rejuvenating the lower face. These structures may be responsible for certain signs of aging, such as the presence of mental creases, an orange-peel appearance, or depression of the lower face [5,6,7]. The mental crease is the deep groove between the lower lip and the prominence of the chin that is produced by contraction of the mentalis muscle. An orange-peel appearance of the chin results from the loss of subcutaneous fat and dermal collagen in the mentum region [8]. These signs occur when the function of the mentalis muscle is not in harmony with the surrounding tissues. Botulinum toxin injections can be used to compensate for these signs by weakening the mentalis muscle.

The application of botulinum toxin in the facial area first began with it being injected into the extraocular muscles for treating strabismus in the 1970s [9]. This toxin weakens the underlying facial muscles to provide satisfactory esthetic outcomes, and represents a novel approach to treating facial rhytides [10]. Botulinum toxin is now commonly used for both therapeutic and esthetic purposes, and its demand is increasing. Botulinum toxin injection is considered to be an ideal cosmetic procedure due to its dramatic effect, less side effects than invasive surgeries, and reversibility of the results [11].

Clinicians who are treating facial rhytides on the lower face need to have a good understanding of the anatomy of the mentalis muscle. Although several methods based on various clinical techniques, anesthesia procedures, and toxin dosages have been proposed for obtaining successful results, no relevant information has been published about the precise morphology and topography of the mentalis muscle.

The aim of this study was to determine the anatomical morphology and location of the mentalis muscle and thereby provide anatomical information for facilitating clinical procedures designed to rejuvenate the lower face.

## 2. Results

### 2.1. Morphology of the Mentalis Muscle

The various morphologies exhibited by the mentalis muscle specimens could be classified into two types. In type A (86.4%, 38 of the 44 cases), the mentalis muscle was dome shaped in three dimensions, and this type could be further divided into two subtypes: type A-1, merging (47.7%, *n* = 21); and type A-2, separate (38.6%, *n* = 17). In type A-1, two bilateral mentalis muscles merged with each other, and there was no space between these two muscles. In type A-2, the two mentalis muscles were separated, and there was space between them (Figure 1A,B).

In type B the mentalis muscle was flat, which constituted 13.6% of the cases (*n* = 6). These mentalis muscles were comprised of only a few muscle fibers, which made the muscle appear quite thin and not three dimensional. The muscles were separated, and they had a trapezoidal shape (Figure 1C).

### 2.2. Location of the Mentalis Muscle

The location and the distribution of the mentalis muscle is listed in Table 1 and illustrated in Figure 2. Medial origin point (MO) was located 4.1 ± 1.8 mm (mean ± SD) lateral to the midline and 20.5 ± 3.7 mm inferior to the horizontal line; the corresponding locations for lateral origin point (LO), medial insertion point (MI), and lateral insertion point (LI) were 13.1 ± 3.4 and 21.3 ± 4.0 mm, 1.8 ± 2.0 and 32.7 ± 4.6 mm, and 10.4 ± 3.2 and 35.1 ± 4.5 mm, respectively. The length and width of the mentalis muscle were 18.0 ± 3.9 and 10.9 ± 3.0 mm, respectively. The distance from the mental foramen was 16.7 ± 3.8 mm. There was no significant difference between the Korean and Thai group, nor the male and female group (*p* value > 0.05).

### 2.3. Depth of the Mentalis Muscle

The depths and thicknesses of the mentalis muscle and surrounding tissues at the reference points are listed in Table 2 and illustrated in Figure 3. The mentalis muscle was 6.7 ± 1.4 mm below the skin and 4.0 ± 1.4 mm thick. The distance between the mentalis muscle and the bone surface, which can be assumed to correspond to the thickness of the submentalis fat, was 1.1 ± 1.0 mm. The mentalis muscle was present between mean depths of 6.7 to 10.6 mm below the skin surface. There were no significant differences in these dimensions between the left and right sides.

## 3. Discussion

It is important to have a good understanding of the three-dimensional morphology of facial expression muscles in various clinical applications, including when performing minimally invasive procedures and surgical procedures. Unlike other skeletal muscles, facial muscles are not surrounded by fascial structures and have tendons that attach to unusual insertions such as soft tissue, facial skin, or other muscles [12]. Facial expression muscles that insert into the same skin location in different directions maintain the balance between the forces exerted by them [4]. Therefore, when injecting botulinum toxin into the facial area so as to affect muscular activity, clinicians should carefully consider the balance between the adjacent muscles and their positional relationships.

Several studies have proposed points for injecting botulinum toxin into the mentalis muscle. Louran (2001) suggested performing superficial medial injections into the upper part of the muscle [13]. Carruthers and Carruthers (2003) suggested that the safe location for injecting into the mentalis muscle was the most-distal point from the orbicularis oris muscle, in order to avoid complications such as weakening that the latter muscle [8]. Klein (2002) pointed out that, according to Mahant et al. (2000), the clinical application of botulinum toxin generally has been safe and well tolerated [14,15]. However, if muscles are adjacent to each other, such as in the facial area, local diffusion can produce unwanted effects in nontargeted muscles. For example, incorrectly injecting botulinum toxin into the mentalis muscle can result in paralysis of adjacent muscles such as the depressor labii inferioris (DLI) or orbicularis oris muscle. Unwanted injection or toxin spreading into the DLI would paralyze it and could result in speaking difficulties, while the same effect of the toxin in the orbicularis oris muscle can cause an incompetent lip or facial asymmetry [10,13].

The mentalis muscle was previously described as dome shaped [16]. However, flat mentalis muscles (type B) constituted 13.6% of the specimens (*n* = 6) in the present study. The muscle fibers of type-B mentalis muscles were quite thin and spread into a wide shape compared to those of type-A mentalis muscles. This implies that more-precise anatomical knowledge is required to accurately target the mentalis muscle when performing botulinum toxin injections.

Three types of the mentalis muscle morphology also can be observed in the ultrasonographic images. In the horizontal ultrasonographic image taken from the point 5 mm above pogonion, three morphological types of the mentalis muscle are well distinguished as shown in Figure 4.

The spatial relationship between the left and right mentalis muscles has been reported previously. Zide and McCarthy (1989) described that the mentalis muscle was separated by a firm septum [3]. Hur et al. (2013) showed that the medial fibers of the mentalis muscle descended anteromedially to cross together on the inner surface in all specimens [2]. The present cadaveric study made observations only of the outer surface of the mentalis muscle, and focused on the presence of space between the two bilateral muscles rather than inner structure of the muscle fibers. We found that the two mentalis muscles were separated and that there was space between these muscles in 17 of the 44 specimens (38.6%, type A-2), with connective tissue and some fat occupying this space.

It is difficult to inspect the location and size of the mentalis muscle from the skin surface due to it frequently being present as a deep facial muscle. There are previous reports on the morphology, direction, and function of the mentalis muscle. The left and right mentalis muscles were found to be separated by a mean distance of 12 mm, and their mean width at the origin was approximately 5 mm [17,18]. However, accurate information about the anatomical location and size of muscle has not been reported previously. The present study measured the origin and inserting points of the mentalis muscle and its mean length and width, as well as its distance from the mental foramen. This study found that in most specimens the mentalis muscle was present at the area 5–10 mm from the facial midsagittal line and 20–30 mm from a horizontal line connecting the mouth corners.

Ultrasound imaging has been widely used in various fields for both diagnosis and treatment. It has the advantages of being easy to use and enables visualization of the internal structures in real time with almost no harmful side effects. In contrast to other sites, the facial area has been rarely analyzed by ultrasound previously due to the presence of thin and multilayered anatomical structures and atypical muscular structures. However, the increasing demand for minimally invasive esthetic procedures has resulted in clinicians beginning to utilize ultrasound technologies for guiding accurate injection procedures, and related studies have also been conducted [19,20]. As for the mentalis muscle, Volk et al. (2013) presented the cross-sectional area of the mentalis muscle through ultrasonographic study [21]. However, since the face area is a multilayered structure, a study on the depth was required to present an ideal injection site.

The present study used ultrasonography to detect and measure the precise location of the mentalis muscle. The measurements were made at the two points 5 mm lateral to the pogonion due to those points (PL and PR) being assumed as safe locations for preventing unintended DLI injection, and also this is where the mentalis muscle is mostly present in the same time. The distance from the skin surface to the mentalis muscle was mostly within the range of 5–9 mm, and the mean thickness of the muscle was 4 mm. This indicates that injecting botulinum toxin at a depth of about 9 mm will accurately target the muscle layer in most cases.

The distance between the mentalis muscle and the bone surface in the mentum area was found to be 1.0 ± 1.0 mm. Thus, this bony surface may be considered as the ideal plane for botulinum toxin injections targeting the mentalis muscle. However, from previous studies it is known that the upper fibers of the mentalis muscle run horizontally toward the anterior skin surface while the lower fibers run inferomedially or vertically, and so deep injections performed close to the bone surface may have greater effects on the lower muscle fibers that travel down vertically [2]. An orange-peel appearance is obviously due to the muscle fibers that travel forward and insert in the anterior skin of the chin, so performing a deep injection close to the bone surface may be insufficient or result in oversuppression of the whole mentalis muscle function or imbalance of contraction inside the muscle. Yu et al. (2020) reported the paradoxical bulging of the chin after the BoNT-A injection in mentalis muscle [22]. This side effect is thought to be due to an imbalance in action of toxins in the muscles, and therefore injections covering all muscle depths were suggested. The imbalance of the mentalis muscle function would result in such esthetic and functional problems, including ptosis of the chin or an incompetent lip, so it is recommended to choose the depth of injection based on the treatment purpose.

As shown in the cadaveric study, mentalis muscles are mostly dome shaped (Type A, 86.4%), but may be flat and comprised of only a few muscle fibers (Type B, 13.6%). Most of the botulinum toxin injection indications will not correspond to type B with few muscle fibers because the purpose of the procedure is to suppress the excessive activity of the mentalis muscle. However, if necessary, clinicians will be able to adjust the dose by checking muscle thickness through the ultrasonographic examination.

The BoNT-A is known to spread 15 to 45 mm from the injection site, and the range depends on the dose and the volume of toxin [23,24]. Single injection into the appropriate location and depth would be sufficient to paralyze the whole mentalis muscle because the length and the width of the mentalis muscle is 18.0 ± 3.9 and 10.9 ± 3.0 mm according to the result of the study. However, since it is known that the area affected by the toxin is mainly determined by the dose and the volume, it is well known that the multiple toxin injection with distributed total dose along the target muscle would help contain the effect of toxin within it [24,25]. Therefore, if the clinician also can accurately inject into the target muscle, multiple point injection can be selected as a method, and it will be safer and more effective with the aid of the ultrasonographic evaluation.

## 4. Conclusions

Precisely locating the mentalis muscle has been difficult due to variations in its morphology and location. The new information reported here may be useful when determining the location of the mentalis muscle in specific patients, including when identifying the most effective and safe botulinum toxin injection point during esthetic procedures for weakened facial rhytides on the lower face.

## 5. Materials and Methods

### 5.1. Morphology of the Mentalis Muscle

Forty-four adult hemifaces from 5 Thai cadavers (4 males and 1 females) and 21 Korean cadavers (14 males and 7 females) with an age range of 43–88 years and a mean age of 69.5 years were used in this study. All of the cadavers had been donated legally to the Yonsei Medical Center (Seoul, Republic of Korea) and Chulalongkorn University (Bangkok, Thailand), and they had no history of trauma or neuromuscular disorder or surgical procedures in the mentum area.

The skin and subcutaneous tissues of the lower face were first removed, and the perioral muscles were then dissected to reveal the origin of the mentalis muscle. A detailed dissection was performed inferiorly from the origin area of the muscle to the skin of the chin, with extreme care being taken not to damage the small fibers of the mentalis muscle. The observed morphology of the mentalis muscle was classified into two types: dome shaped and flat. The dome-shaped muscles were further subdivided into two subtypes: merging and separated (Figure 5).

### 5.2. Location of the Mentalis Muscle

The location and size of the mentalis muscle were measured using digital calipers (CD-15CP, Mitutoyo, Kawasaki, Japan). The facial midsagittal line and a horizontal line connecting the mouth corners were used as reference lines (Figure 6). Four points of the mentalis muscle were identified: MO, LO, MI, and LI. The following parameters were measured:MO location.LO location.MI location.LI location.Length of the mentalis muscle.Width of the mentalis muscle.Distance between the mental foramen and the lateral margin of the mentalis muscle.

### 5.3. Depth of the Mentalis Muscle

Sixty-six adult hemifaces from 33 healthy young Korean volunteers were included in this ultrasonographic study (20 males and 13 females, mean age 23.6 years). The most-anterior midsagittal point on the contour of the chin (pogonion) was marked before obtaining an image using a real-time two-dimensional B-mode US device with a high-frequency (15 MHz) linear transducer (E-CUBE 15 Platinum, ALPINION Medical Systems, Seoul, Korea) with the volunteer in the semisupine position. A linear probe was applied horizontally at the pogonion perpendicular to the skin surface to obtain the ultrasound image (Figure 7A).

The measurements were made at two points: 5 mm left (PL) and 5 mm right (PR) of the pogonion. Special efforts were made to prevent any pressure from the ultrasound probe distorting the soft tissue. The depth of the mentalis muscle below the skin surface (corresponding to the combined thickness of the skin and subcutaneous tissue), the thickness of the mentalis muscle, and the distance from the bone to the mentalis muscle (corresponding to the thickness of the submentalis fat) were measured using the ImageJ program (National Institutes of Health, Bethesda, MD, USA) (Figure 7B). All of the study procedures were approved by the institutional review board of the Yonsei University Dental Hospital (IRB No. 2-2017-0023), and they were fully explained to all volunteers, who then provided written consents.

## Figures and Tables

**Figure 1 toxins-13-00096-f001:**
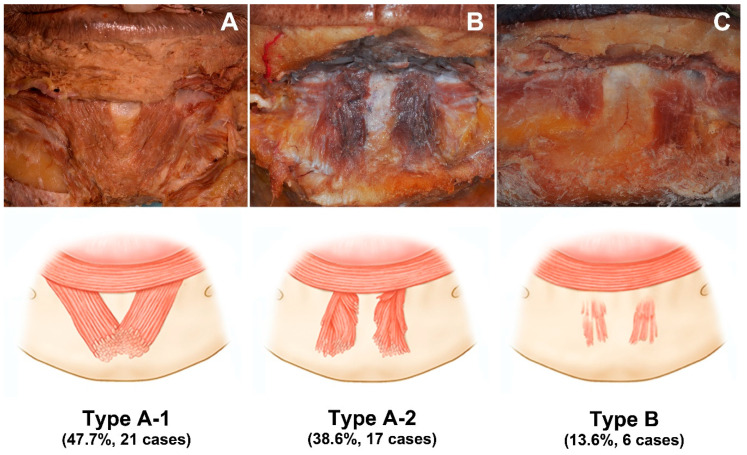
Three types of mentalis muscle morphology. (**A**) Type A-1, two bilateral mentalis muscles merged with each other. (**B**) Type A-2, the two mentalis muscles were separated. (**C**) Type B, mentalis muscle was flat and comprised of only a few muscle fibers.

**Figure 2 toxins-13-00096-f002:**
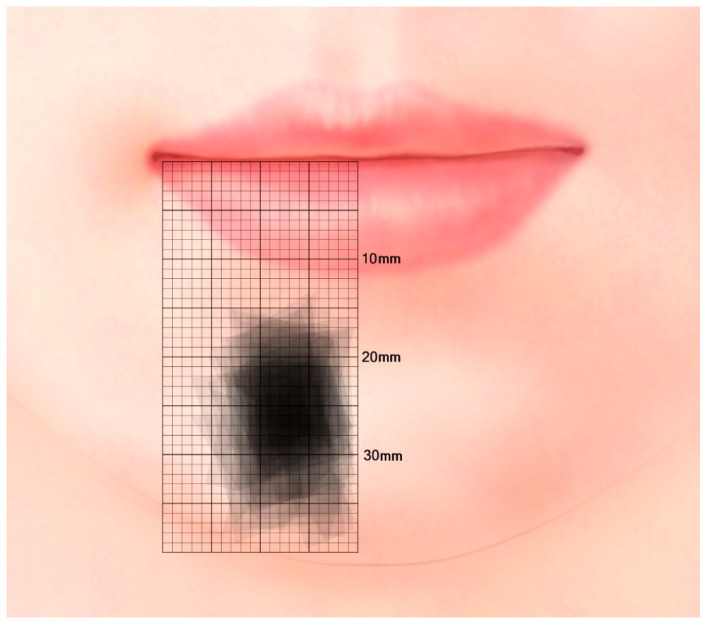
Distribution of the mentalis muscle relative to the facial midsagittal line and a horizontal line connecting the mouth corners. In most specimens the mentalis muscle was present at the area 5–10 mm from the facial midsagittal line and 20–30 mm from a horizontal line connecting the mouth corners.

**Figure 3 toxins-13-00096-f003:**
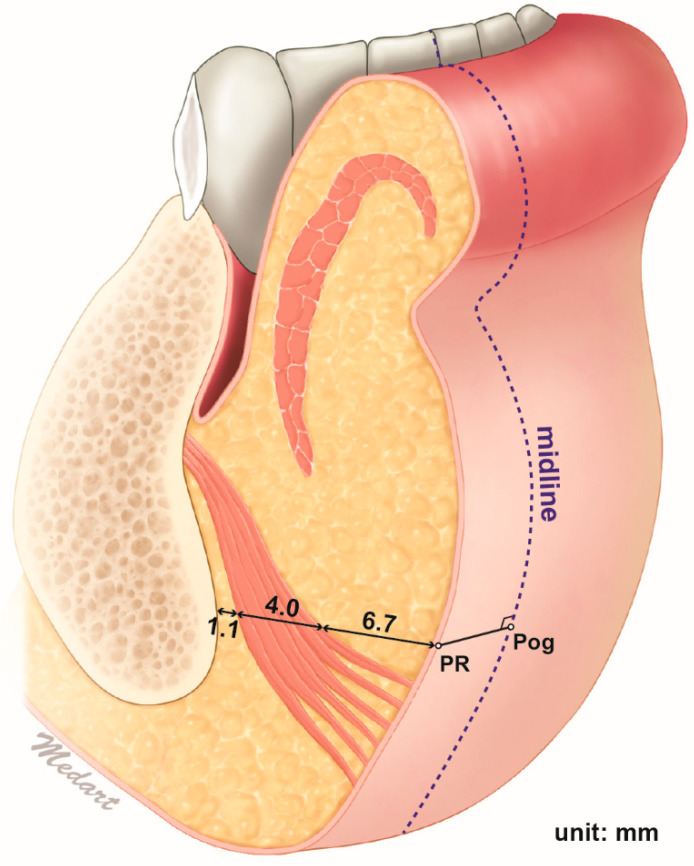
*Z*-axis anatomy of the mentalis muscle. The depth of the mentalis muscles was measured at the point 5 mm left and right from the pogonion. The depth of the mentalis muscle below the skin surface, the thickness of the mentalis muscle, and the distance from the bone to the mentalis muscle were 6.7 ± 1.4, 4.0 ± 1.4, and 1.1 ± 1.0 mm, respectively. The mentalis muscle was present between mean depths of 6.7 to 10.7 mm below the skin surface.

**Figure 4 toxins-13-00096-f004:**
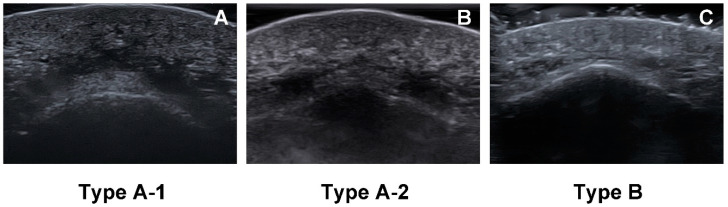
Three types of mentalis muscle morphology observed in ultrasonographic images. B mode, transverse view, 15-MHz linear transducer. (**A**) In type A-1, the mentalis muscle is dome shaped, and the muscle fibers on each side are merged. (**B**) In type A-2, the mentalis muscle is dome shaped, but the muscle fibers on each side are separate. (**C**) In type B, the mentalis muscle was flat and comprised of thin fibers.

**Figure 5 toxins-13-00096-f005:**
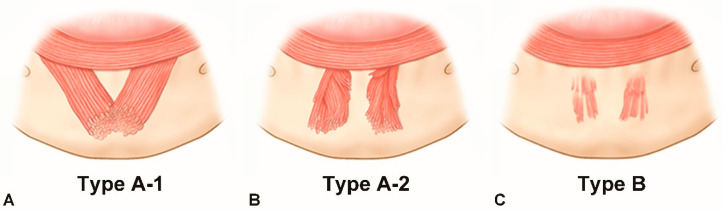
Images and illustrations demonstrating various morphologies of the mentalis muscle. Some of the muscle specimens were dome shaped, where the muscle fibers on each side are merged (**A**) or are separate (**B**). Other muscle specimens were flat and comprised of thin fibers (**C**).

**Figure 6 toxins-13-00096-f006:**
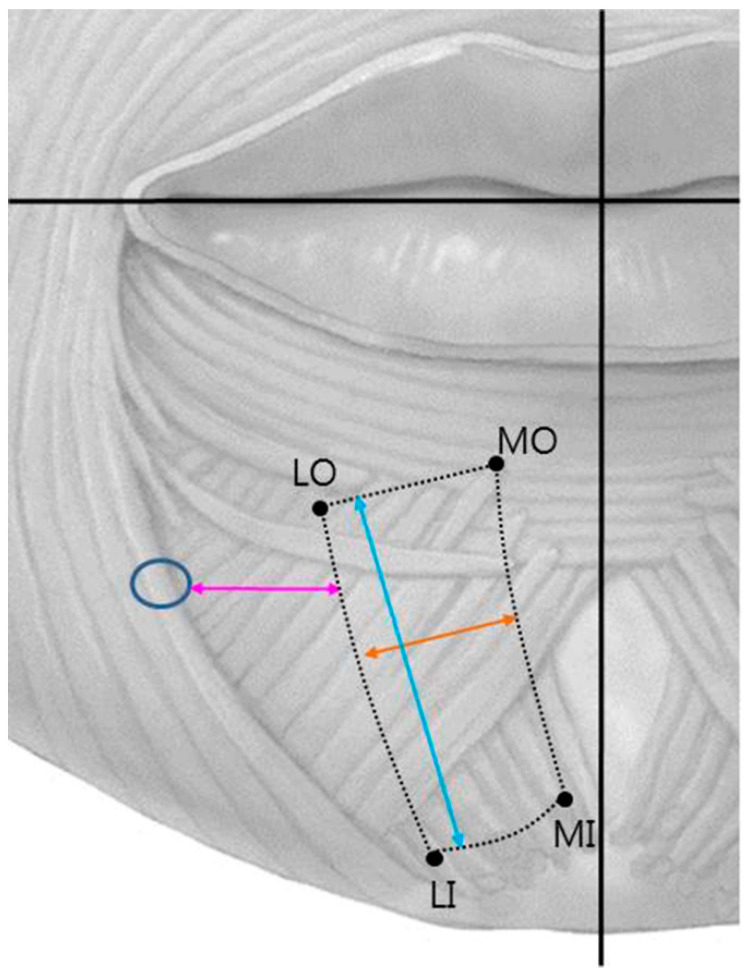
Reference lines and measurement points. MO, medial origin point; LO, lateral origin point; MI, medial insertion point; LI, lateral insertion point; blue circle, mental foramen; yellow arrow, width of the mentalis muscle; blue arrow, length of the mentalis muscle; pink arrow, distance between the mental foramen and the lateral margin of the mentalis muscle.

**Figure 7 toxins-13-00096-f007:**
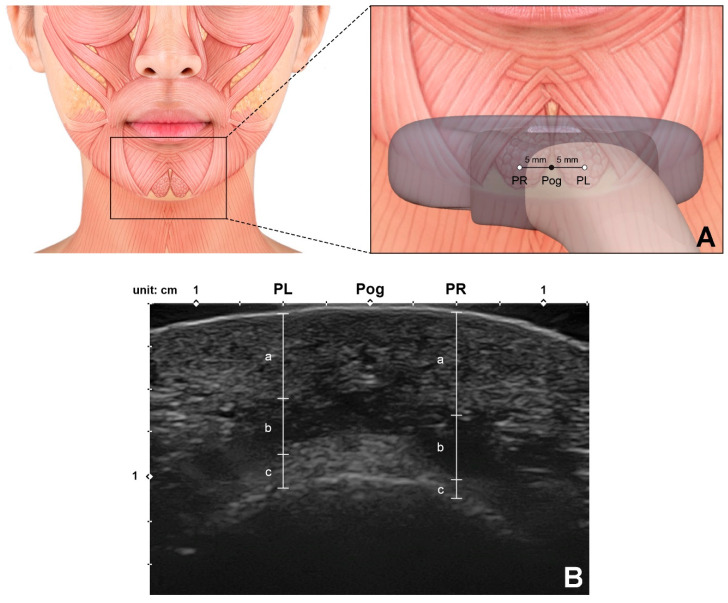
Ultrasonographic analysis of the mentalis muscle. (**A**) Reference points for observing and measuring the thickness of the mentalis muscle. Pog, pogonion; PL, the point 5 mm left from the Pog; PR, the point 5 mm right from the Pog. A linear probe was applied horizontally at the Pog perpendicular to the skin surface to obtain the ultrasonographic image. (**B**) Ultrasonographic image and the parameters measured for the mentalis muscle and surrounding tissues (B mode, transverse view, 15-MHz linear transducer). Each parameter was measured at both PL and PR.; a, the depth of the mentalis muscle below the skin surface; b, the thickness of the mentalis muscle; c, the distance from the bone to the mentalis muscle were measured.

**Table 1 toxins-13-00096-t001:** Location and size of the mentalis muscle. Data are mean ± SD values (millimeters).

	Distance from Midline	Distance from Horizontal Line
Point MO	4.1 ± 1.8	20.5 ± 3.7
Point LO	13.1 ± 3.4	21.3 ± 4.0
Point MI	1.8 ± 2.0	32.7 ± 4.6
Point LI	10.4 ± 3.2	35.1 ± 4.5
Length	18.0 ± 3.9	
Width	10.9 ± 3.0	
Distance from mental foramen	16.7 ± 3.8	

MO, medial origin; LO, lateral origin; MI, medial insertion; LI, lateral insertion.

**Table 2 toxins-13-00096-t002:** Depth of the mentalis muscle. Data are mean±SD values (millimeters).

	Depth Below Skin Surface	Muscle Thickness	Distance from Bone
Left	6.7 ± 1.4	3.9 ± 1.4	1.0 ± 1.1
Right	6.6 ± 1.4	4.0 ± 1.3	1.0 ± 0.9
Total	6.6 ± 1.4	3.9 ± 1.4	1.0 ± 1.0

## Data Availability

The data presented in this study are available on request from the corresponding author.

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
