# Peer review of "Effective Locations for Injecting Botulinum Toxin into the Mentalis Muscle; Cadaveric and Ultrasonographic Study"

_toxins, 2021, doi:10.3390/toxins13020096_

Round 1

Reviewer 1 Report

This is an interesting study assessing the anatomic size and shape of the mentalis muscle with respect to an optimized botulinum neurotoxin (BoNT) application, using postmortem anatomical studies and ultrasound imaging studies. The paper is well written, the graphical illustrations are of high quality, and the methods are sound. There are however several aspects that should be described and discussed in more detail.

  1. Methods section: Please add information on whether a history of neuromuscular disorders (including peripheral facial nerve palsy) had been excluded in the individuals of whom the postmortem specimen were studied.
  2. Methods section: Please provide information which ultrasound probe and which ultrasound frequency was used for visualizing the mentalis muscle.
  3. Methods section: Please describe shortly how you could discriminate the depressor labii inferioris (DLI) muscle from the mentalis muscle on ultrasound in the more cranial transection of the chin. Please add a figure showing an axial ultrasound image of the mentalis and DLI muscle at the position where both, LO and MO, points of mentalis muscle were assessed. Please denote the LO and MO points in this image.
  4. Results: Three different types of mentalis muscle anatomy are identified as a result of the cadaver study, however, this classification was not applied (or not found?) in the ultrasound study. Please explain this discrepancy if present. Otherwise, please provide information on whether with ultrasound at least the types A-1 and A-2 could be differentiated (type B may be indistinguishable from type A-2 on ultrasound). If available, please add the number of cases of each type found with ultrasound.
  5. Discussion: It would be of interest whether the authors consider differential BoNT doses depending on the thickness and shape of the mentalis muscle.
  6. Discussion: The study of Volk GF et al. (Muscle Nerve 2013;47:878-883) should be referred to, who were the first to study the mentalis muscle with ultrasound systematically in healthy subjects, and who have shown that its size (cross-sectional area) is independent from age or height of a person.
  7. Discussion: The statement “In contrast to other sites, the facial area has not been analyzed by ultrasound previously due to the presence of multilayered thin anatomical structures and atypical muscular structures.” should be modified since meanwhile a number of studies (especially from the group of Volk GF and his group) were published. Still, it is true that in the context of BoNT application the number of studies in the facial region is low.
  8. Discussion: The authors may consider including a recent case report (Yu N et al. J Cosmet Dermatol 2020;19:1290-1293) highlighting the use of ultrasound prior to, or during, BoNT injection. The authors might compare the proposed injection points in that previous report with the injection sites suggested with the present findings.

Author Response

//Thank you for your kind and thoughtful reviews.

The attached file has two parts: the first part is the response to reviewer's comment and the second part is the modified manuscript.

Please check the later part of the attached file to confirm the changes in the manuscript.//

Reviewer 2 Report

This study has looked into the mentalis muscle to determine the anatomical morphology and localization via two methods of cadaver inspection and ultrasonography. Determining the location of the mentalis muscle is valuable because it can facilitate proper injection for lower face rejuvenating or other purposes. The authors use 44 hemifaces from Thai and Korean adults’ cadavers. For the ultrasonographic investigation, 66 hemifaces from 33 healthy young subjects were included. Measurements included the depth of the mentalis muscle below the skin surface, the thickness of the mentalis muscle, and the distance from the bone to the mentalis muscle at the two points that were 5 mm 12 lateral to the most-prominent point of the chin. Authors, based on the findings, classified the mentalis muscle into two types based to its shape: type A (86.4%, 38 of the 44 cases) as the dome shaped in three dimensions; and type B (13.6%, 6 of the 44 cases) as flat. The mentalis muscle was present between depths of 6.7 to 10.7 mm below the skin. This paper adds new information about the location of the mentalis muscle that can be applied in the identifying of the injection location and depths during esthetic procedures. This can be for example for weakened facial rhytides on the lower face. There are few points that authors are encouraged to clarify in the revised version:

  • Did the author received approval from the local ethics committee? If so, please add the information in the method section.
  • Did the author find any difference between the Korean and Thai or between the male and males?
  • Please add the rationale for using both methods for the measures in this study. i.e., How these two methods can supplement each other and provide more information.
  • Considering the group of participants, could authors generalize the finding? For example, for other face types in addition to Thai and Korean?
  • Do authors recommend a special needle type or size to be used for mentalis muscle considering their findings?
  • Would that be better for multiple injections or a single injection is recommended considering the findings and about anatomy, morphology of the macule?
  • What can go wrong if the injection site is not determined properly?

Author Response

//Thank you for your kind and thoughtful reviews.

The attachment has two parts: the first part is the response to reviewer's comment and the second part is the modified manuscript.

Please check the later part of the attached file to confirm the changes in the manuscript.//

Round 2

Reviewer 1 Report

The authors have adequately addressed the issues raised by the reviewer.

I recommend the following minor revision:

Please add the 3 ultrasound images included in the cover letter (in the author reply to reviewer's Q4), which represent the 3 shapes of mentalis muscle (A-1, A-2, B), also in the manuscript (e.g. integrated in figure 4 or 6, or in a separate new figure).   

Author Response

/Please see the attachment/

Thank you for your kind and thoughtful review.
